# Landscape Performance for Coordinated Development of Rural Communities & Small-Towns Based on "Ecological Priority and All-Area Integrated Development": Six Case Studies in East China's Zhejiang Province

**Tiezheng Zhao** [1] , **Yang Zhao** [1] **and Ming-Han Li** [2,*]

[1] School of Ecological Technology and Engineering, Shanghai Institute of Technology, Shanghai 200234, China

[2] School of Planning, Design and Construction, Michigan State University, East Lansing, Michigan, MI 48824, USA

* Correspondence: minghan@msu.edu

**Abstract:** Over the last decade, the Chinese government has focused on addressing development challenges in rural areas. The "Ecological Priority and All-Area Integrated Development" concept was thus developed, and it was found to be crucial for rural areas in Eastern Zhejiang Province. A new comprehensive evaluation system was composed by comparing and synthesizing existing Chinese assessment criteria, and landscape performance metrics developed by the Landscape Architecture Foundation. Analytic Hierarchy Process (AHP) and Fuzzy Comprehensive Evaluation Method (FCE) were used to conduct post-development evaluation on six cases using the new evaluation system. The results of four cases show that ecology should be considered a high priority when dealing with rural community and small town developments. The other two cases emphasizing infrastructure development verified that "coordinating the development of rural communities and small town area" is crucial for building sustainable and livable rural communities, and avoiding redundancy and inefficiency. The newly developed comprehensive evaluation system integrates existing systems with a broader vision and is more holistic in its objectives for the region. The development-led intervention (based on landscape performance evaluation) is conducive to the implementation of a more scientific and comprehensive development model, with predictable performance.

**Keywords:** ecological priority, All-Area Integrated Development; coordinated development of rural communities & small towns; landscape performance evaluation; rural landscape architecture

## 1. Introduction

### 1.1. Research Background

#### 1.1.1. The Focus of Rural Development in the Current China Context

For decades, the Chinese government has led a series of development schemes in rural areas to meet the challenges of eco-environment destruction, population outflow and aging, decline of traditional industries, and rising demand for regional tourism services. A rural revitalization process that emphasizes the development of rural area was integrated with traditional urban development strategies.

1.1.2. Policy Guidance on Rural Construction Development of China over the Last Decade

The Chinese government began to emphasize rural construction in 2002. In 2014, the "New Urbanization" idea was introduced to orient the development of rural areas at the macro level to deal with the developmental inequalities between urban and rural areas, especially in areas of environmental management, infrastructure, public services, etc. [1]. In 2015, the "Construction of Beautiful Rural Villages" was carried out [2]. In 2018, the "Rural Community Environment Improvement" scheme was posted [3], focusing on the improvement of rural living environment, followed by the "Rural Revitalization" program, which stressed on the comprehensive development of rural areas [4]. Such rural policies evolution indicates that the guideline has evolved from focusing on rural material production to a comprehensive system of rural industry, living and ecology integration.

Because of the different regulating departments, the above rural policies focus on different concerns for different related topics. The basic contents seek to address the resolution of rural life, development of rural industry, and improvement of rural human settlements. However, systematic guidance for rural eco-environment protection, environment resource sharing, and all-area development coordination has been lacking. Under the new requirements of rural construction, it is imperative to propose the new comprehensive evaluation system. This new comprehensive evaluation framework can address more inclusive assessment of post-development and effective adoption in the Chinese region.

1.1.3. Research Brief on Rural Development of China over the Last Decade

Chinese researchers Wang Zhuyun and Chang Jiangju classified China's rural construction research system as economic and industrial research, village construction and planning research, and traditional culture and local culture research. The imminence of China's rural construction was summarized in four aspects: Imbalanced regional development, imperfect rural infrastructure, the serious problem of rural community hollowing, and insufficient rural development momentum [5].

In a research work by Chinese researchers Yao Long and Liu Yuting, China's rural development model was summarized from a different level: From the perspective of development momentum factors, China's rural development model can be classified as exogenous and endogenous. In terms of space system, China's rural development model is mainly a cluster development model. In terms of regional features, China's rural areas like Wenzhou of Zhejiang, South of Jiangsu, the Pearl River Area of Guangdong, and some other places share a more common rural village-style development model. However, China's rural development model has not sufficiently integrated with rural planning. Researchers Yao Long and Liu Yuting have suggested that further study follow the spatial and regional lead [6].

## 2. Case Study Overview

*2.1. Overview of the Cases*

Since 2008, the research team has led and participated in a large number of urban and rural planning, planning and construction projects, and long-term local planning management work, most of which happened in the Siming Mountainous area of eastern Zhejiang. Through planning and construction practices, the research team obtained comprehensive environmental improvement action plans, master plans, tourism development plans, village plans, small towns plans, and regional collaborative development planning, etc., covering more than ten villages and six townships (as shown in Figure 1).

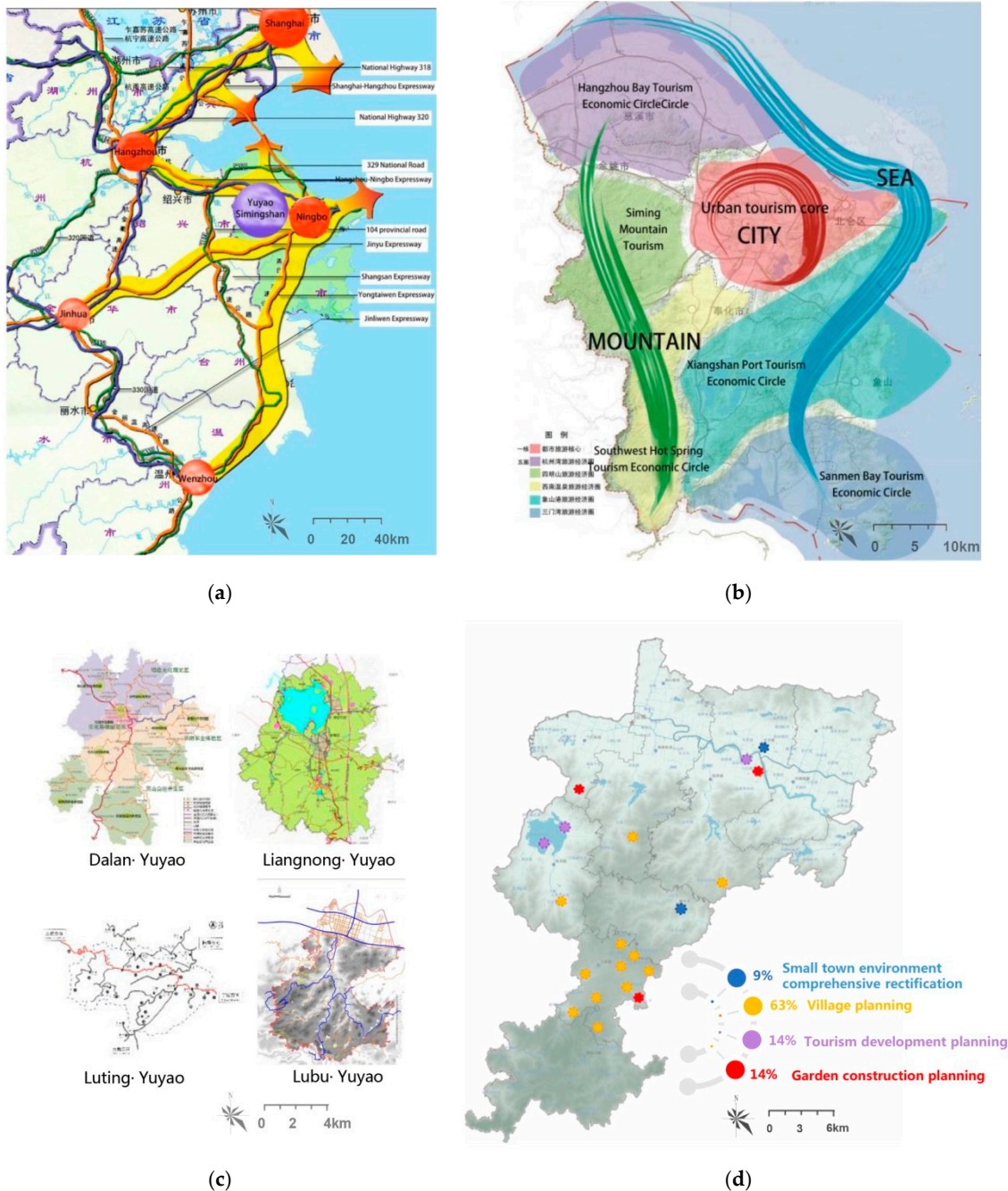

**Figure 1.** Research cases maps. (**a**) The relationship between Yuyao Siming Mountainous Area in the Yangtze River Delta, (**b**) The relationship between Yuyao Siming Mountainous Area with Yuyao city and East China Seashore, (**c**) Map of key towns: Dalan/Liangnong/Luting/Lubu, (**d**) Village & town sample cases distribution map (Yuyao, Siming Mountainous Area).

In this research, six typical cases from Dalan, Luting, Liangnong, and Lubu were selected from among a large number of cases for analysis. They cover the key locations of the overall development in the region with exemplary characteristics such as ecological advantages, livable rural residential community compatibility, regional rural tourism development booming, rural and small-town infrastructure sharing, etc. The construction cases are mainly involved in village planning cases, garden construction planning cases, tourism planning cases, small town environment comprehensive remediation cases, etc. Most of the landscape construction and infrastructure implementation has already been completed. Assessing their post-development performance will

provide a holistic representation of the "Construction of Beautiful Rural Villages" Policy and the impact of its strategic vision.

## 2.2. The Coordinated Development of Villages and Towns from the Perspective of All-Area Integrated Development

The concept of ecological priority indicates that development should be aimed at protecting the environment and resources and conducting a circular economy development mode. The concept of all-area integrated development refers to promoting regionally balanced development through resource-sharing and coordination of symbiosis. In a 2017 work report, the Chinese government stressed on the following: "To give full play to the role of the city in leading rural areas and the role of rural areas in promoting urban development" [4]. In July 2016, the Zhejiang Provincial Government proposed that "ecological priority should be integrated throughout the region and the development of the Qiantang River should be comprehensively promoted." [7]

The urban-rural system structure of "City → Central town → General town → Central village → Natural village" that has lasted for a long time in China is a top-down chain-like, single-cycle development pattern of villages and towns. The resources acquired by cities from rural areas are more than the city's supply of products/services to rural areas (as shown in Figures 2 and 3). The resources of each village and town are divergent based on independent development, and the benefits are independent due to dispersed unrelated villages and towns (as shown in Figure 4) [8].

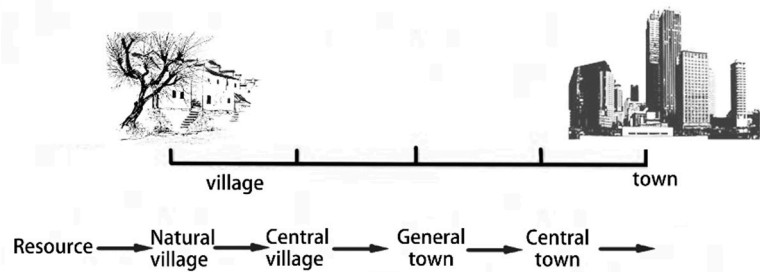

**Figure 2.** Chain-like, single-cycle development model of villages and towns.

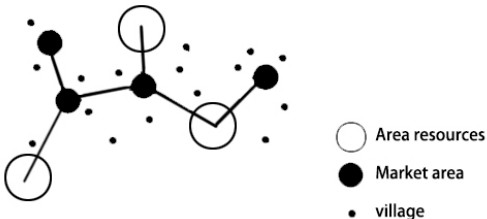

**Figure 3.** Independently dispersed village and town development benefits pattern.

The coordinated development of villages and towns from the perspective of "Ecological Priority and All-Area Integrated Development" represents a new form in China with a network-like pattern. The area townships, central administrative villages, and natural villages participate in the entire cycle as a node in this network system (Figure 5). Through resource-sharing and coordination of symbiosis, mutually beneficial symbiotic development of villages and towns is achieved. The comprehensive benefits are far greater than the former scattered single-chain type, which intensified resource consumption [8].

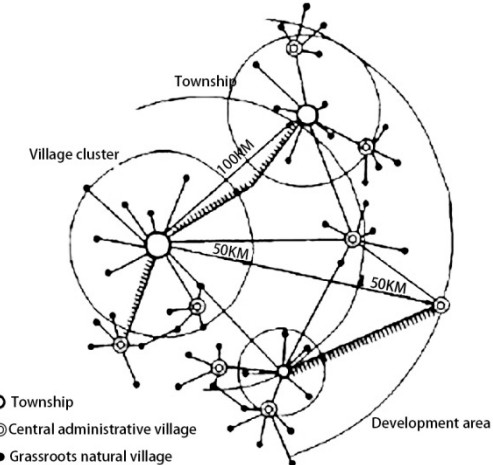

**Figure 4.** Reciprocal symbiosis and synergy of village and town development pattern.

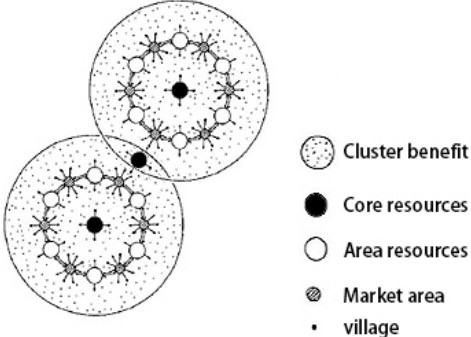

**Figure 5.** Multi-level, village-town synergy development model.

## 2.3. Coordinated Development of Villages and Towns in the Siming Mountainous Area of Yuyao, Zhejiang

The development area of the villages and towns in Siming Mountainous Area is not the typical small towns and villages scattered type. These administrative villages cover natural villages and townships with beautiful rural landscape features. It is closely intertwined with many developmental aspects of rural ecology, industry, and livability. The mountain, river, field, forest, and lake systems embodying the characteristics of beautiful rural areas are integrated into one. On this basis, the conceptual boundary of villages and towns is blurred, and the village-city synergy module is the most basic development control unit. This rural revitalization model of villages and towns can fully echo the new establishment of the national all-area land use planning, which includes the belt system of village modules and continuous village modules. In our opinion, this all-area integrated collaborative development model can give full play to achieve resources and infrastructure-sharing for peripheral towns, promoting regional linkage between various towns (as shown in Figure 6).

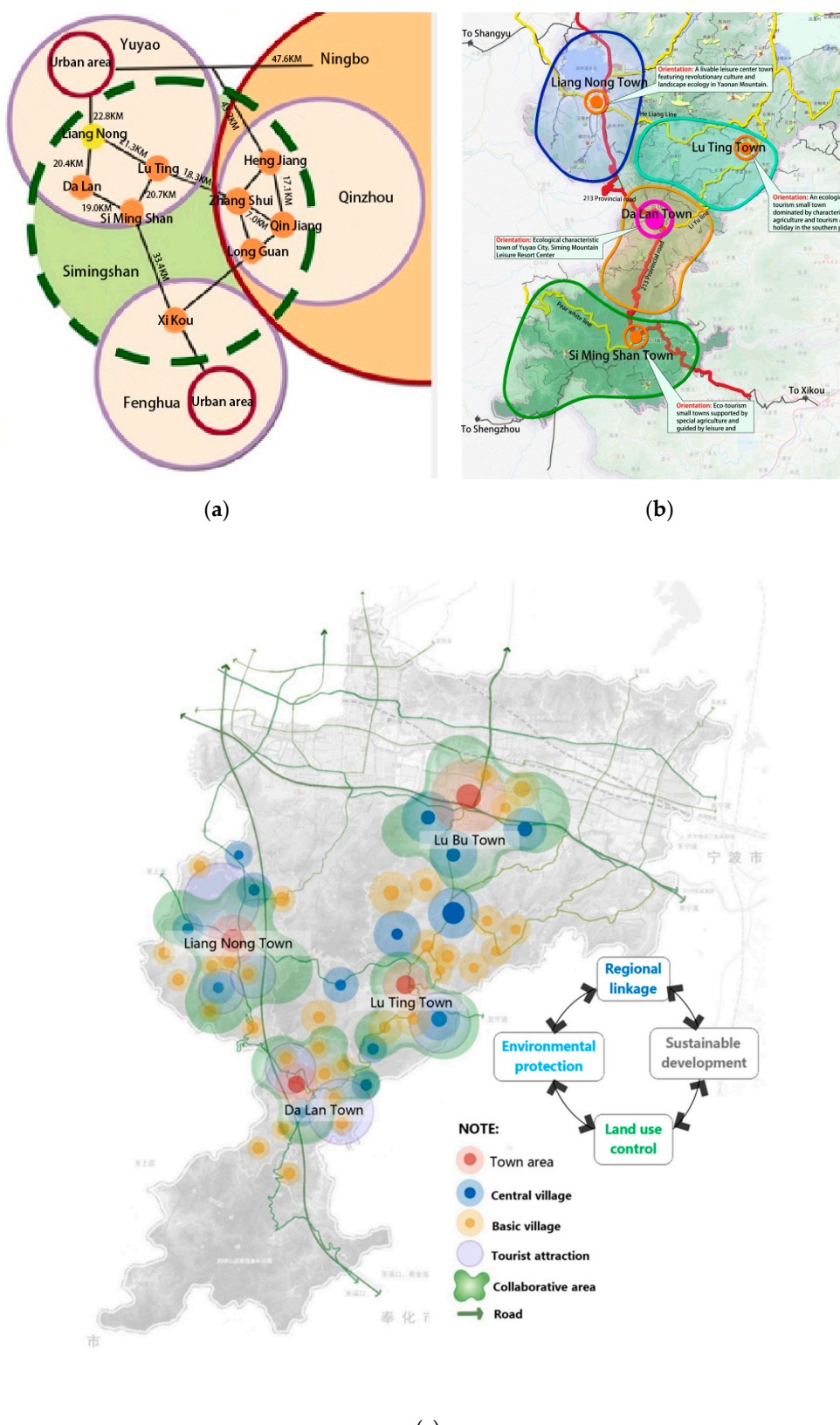

(**a**)

(**b**)

(**c**)

**Figure 6.** Coordinated development model of villages and towns in Siming Mountainous Area, Zhejiang. (**a**) Geographic relationship between Yuyao, Ningbo, Fenghua, and Siming Mountainous Area, (**b**) The Regional Relations of Liang Nong, Lu Ting, Dalan, (**c**) Schematic diagram of coordinated development of the four towns: Dalan, Luting, Liangong and Lujing.

## 3. Methodology

### 3.1. Introduction of Research Methods

3.1.1. Analytic Hierarchy Process (AHP)

Analytic hierarchy process (AHP) is a qualitative and quantitative combination, systematic and hierarchical analysis method proposed by American operations researcher Thomas L Saaty and Professor Sadie of the University of Pittsburgh in the early 1970s [9]. It is often used to evaluate the assignment weight of index weights in the system. It can greatly reduce the disadvantages of subjective assumptions (as shown in Figure 7).

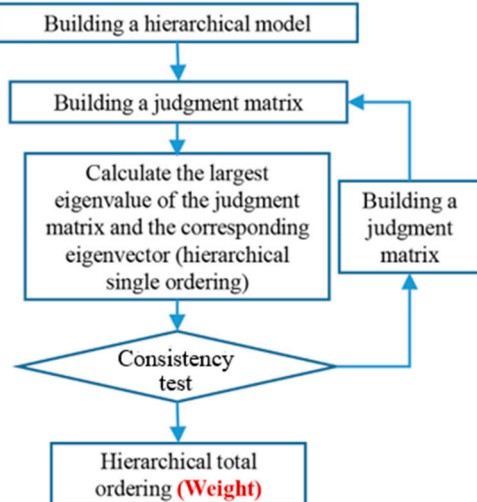

**Figure 7.** Analytic Hierarchy Process steps.

3.1.2. Fuzzy Comprehensive Evaluation Method (FCE)

Fuzzy comprehensive evaluation method is a semi-quantitative and semi-qualitative analysis of multi-factor events that are not suitable for quantification [10]. This comprehensive evaluation method is suitable for solving various non-deterministic problems.

The research team adopted both the AHP and FCE methods to conduct initial regional evaluation and verify the new comprehensive evaluation system based on "Ecological Priority and All-Area Integrated Development" (as shown in Figure 8).

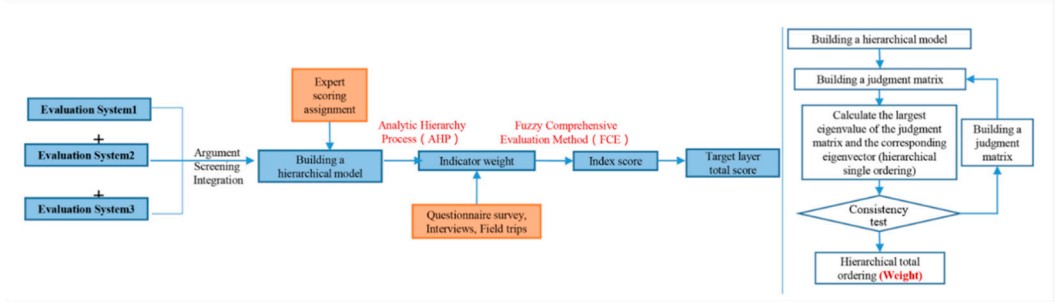

**Figure 8.** Steps of comprehensive evaluation.

### 3.2. Research Ideas and Methods

The main research objectives include two parts (Figure 9): (1) Discussion of a new comprehensive evaluation system, and (2) verification of the new system using case studies. By comparing the existing evaluation indicators of different levels and different modules, the new comprehensive

rural post-planning evaluation system is formed, based on "Ecological Priority and All-Area Integrated Development".

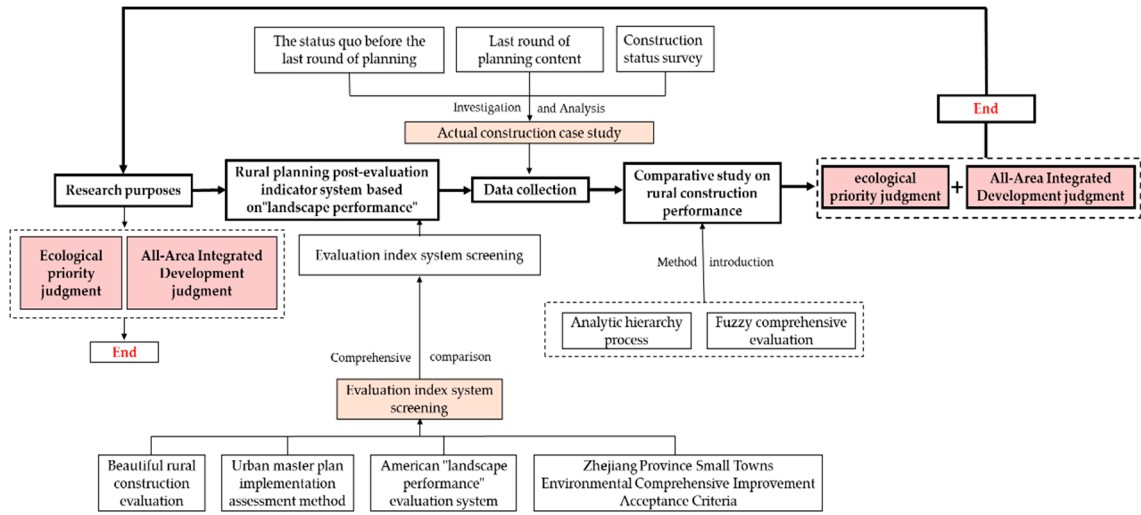

**Figure 9.** Research process and framework.

For the case study, the author selected different planning and construction projects around Siming Mountainous Area and summarized the pre-planning status, latest planning contents, and planning and construction status. The rural planning post-construction evaluation system based on "Ecological Priority and All-Area Integrated Development" is constructed, combined with analytic hierarchy process (AHP) [9] and fuzzy comprehensive evaluation (FCE) method [10] to conduct the cases study. Six cases were selected to conduct empirical research on rural construction performance and to demonstrate how to further promote ecological priority and all-area integrated development, and coordinate development in regional rural communities and small towns.

### 3.3. Development of the New Comprehensive Evaluation System

#### 3.3.1. Overview of Existing Evaluation Systems

The Chinese government has issued guidelines for the development, planning and construction of villages, towns and small towns for a decade now. The three selected representative evaluation guidelines systems are: The 2009 "Measures for the Implementation of the Urban Master Plan Implementation (for Trial Implementation)" [11], the 2015 "Guidelines for the Construction of Beautiful Villages" [2], and the 2016 "Three-Year Action Plan for Comprehensive Environmental Improvement of Small Towns in Zhejiang Province" [7].

The introduction of these evaluation criteria has helped village and towns administrators offer strategic guidance for future development. However, each evaluation system is for a single type of planning and construction project. In the context of the new stage of land and space planning, it is imperative to evaluate the construction results of various stages covering different types of construction projects timely and effectively.

Internationally, there is the UK Annual Monitoring Report (AMR) issued by the Greater London Authority in 2004 [12], the "CASBEE-City" Evaluation System issued by the Comprehensive Environmental Assessment Committee of Japan [13], and the Landscape Performance Series released by the Landscape Architecture Foundation in the U.S. [14]. This study selected the Landscape Performance Series evaluation system as a key reference to compose the new comprehensive performance evaluation framework for systematic focus on ecological, economic, and social benefit assessment [15] (Table 1).

**Table 1.** Comparison of different evaluation systems.

| Name | Year of Announcement | Issued by | Evaluation Object Range | Main Evaluation Content | Main Evaluation Method Used by Indicators | Advantages | Disadvantages |
|---|---|---|---|---|---|---|---|
| Urban master plan implementation assessment method | 2009 | Housing and urban-rural development of the People's Republic of China | City, built-town area | Implementation of planning objectives and implementation of mandatory content such as land use, transportation, industry, and environmental protection | 1. Before and after planning data comparison of various indicators. 2. Site field inspection verification | 1. It is a statutory planning assessment and is authoritative. 2. There are many mandatory contents, and the evaluation coverage is wide, including social and economic aspects | 1. Weak cooperation between villages development. |
| Zhejiang Province Small Towns Environmental Comprehensive Improvement Assessment Score | 2016 | Zhejiang Small Town Environmental Comprehensive Improvement Action Leading Group Office | Small towns in Zhejiang Province | External image remediation as the core, including environmental sanitation, urban order, township and town appearance nodes, etc. | Field research, on-site inspection | The evaluation content is rich and fits the town construction, and it is highly targeted to the villages and towns in this research area | Indicators are too targeted to quantify, weak versatility |
| Beautiful rural construction evaluation index | 2018 | State General Administration of the People's Republic of China for Quality Supervision and Inspection and Quarantine and the National Standards Committee | Beautiful country building | Village construction, ecological environment, economic development, public service content, and software aspects such as facility management, maintenance, funding, staffing, etc. | 1.Evaluation of the report 2. field investigation 3. access to on-site information | 1. Part of the indicators can be quantified and accurate 2. The performance evaluation of rural construction is highly targeted and conforms to the actual situation of rural China | The evaluation of social development performance such as population is weak |
| US "landscape performance" evaluation index | 2010 | Landscape Architecture Foundation | Completed urban park green space | Ecological performance evaluation, including land, water, habitat, carbon, energy and air quality, materials and waste, economy [16] | 1. Information Verification 2. Field investigation | 1. In line with the requirements of the era of sustainable development 2. Has a quantifiable calculation toolbox [17,18] | 1. The calculation toolbox need adaptation for practical use in China 2. The content of the evaluation indicators is too microscopic for the performance of village construction [17,18] |

### 3.3.2. Building of New Comprehensive Evaluation Systems

The research team developed the new comprehensive evaluation system by first reviewing the four existing evaluation systems: "Landscape performance" of the Landscape Architecture Foundation (LAF) in the United States [14], "The Evaluation Method of Urban Master Plan (2009), China" [11]. "Beautiful Rural Construction Evaluation (2018), China" [2], and "Three-Year Action Plan for Comprehensive Environmental Improvement of Small Towns in Zhejiang Province, China" [3].

The "Urban Master Plan Implementation Assessment Method" [11] adopts a combination of qualitative and quantitative methods, in a bid to carry out the evaluation by comparing the data of various indicators and field inspection and verification. Due to differences in the scale of the assessment objects, the research team excluded specific indicators that are not applicable to the rural construction scenarios of its original assessment system, such as urbanization rate, intercity railways, waterways, etc.

"Interim Measures for the Assessment and Acceptance of Environmental Comprehensive Improvement Actions in Small Towns in Zhejiang Province" [7] mainly aims at the evaluation of small-town construction in Zhejiang Province. The plan is committed to improving the appearance of the town and functions based on the township's characteristics. The evaluation methods are mainly on-site inspections, and the external appearance remediation is a key factor. Other factors include environmental sanitation, townscape, and township nodes construction. The assessment index is highly targeted to villages and towns in the Siming Mountainous area, especially small towns.

"Beautiful Rural Construction Evaluation (2018), China" [2] was the newest evaluation ordinance. It covers the assessment for both physical construction and plan implementation (such as maintenance, funding, and related staffing).

"Landscape Performance" of LAF [9] mainly aims to evaluate the performance of the operational status of completed projects. According to LAF, the system covers performance categories such as land, water, habitats, carbon, energy and air quality, materials and garbage, economy, society, etc. This system has been implemented into 100 exemplary built projects to quantify environmental, economic and social benefits. It is the collaboration work of designers and/or academic research teams to collect evidence of landscape performance. It is also by far the most comprehensive metrics that has been used to assess projects on different scales—from site improvement to urban planning.

Based on the review and analysis of these four evaluation systems mentioned above, the new comprehensive evaluation system includes first-level indicators (economic performance, social performance, and ecological performance [19]) from the "landscape performance" evaluation system as the key performance framework because of its simplicity and clarity to holistically summarize landscape performance. Selected second-level and third-level indicators from the three existing China evaluation systems were included to cover the characteristics of the regional environment. Table 2 shows the new comprehensive evaluation system under the guidance of "Ecological Priority and All-area Integrated Development" (Table 2).

**Table 2.** Post-construction evaluation system for Coordinated Development of Rural Communities and Small Towns Based on "Ecological Priority and All-Area Integrated Development".

| First Level | Second Level | Third Level | Remark | Source |
|---|---|---|---|---|
| Economic performance | Industrial economy | Economic strength | Gross regional product and per capita output value | Government statistics |
| | | Industrial structure | Tertiary industry value and proportion | |
| | | Visitor volume | | Statistical report |
| | | Resident employment type | General type of employment | Questionnaire |
| | | Parking revenue | Tourism is more suitable | |
| | Social development | Urbanization rate | For small towns evaluation | Government statistics |
| | | Aging rate | | |
| | | Rural hollowing percentage | | |
| | | Population growth rate | | |

**Table 2.** *Cont.*

| First Level | Second Level | Third Level | Remark | Source |
|---|---|---|---|---|
| | Space usage | Residential space distribution | Planning implementation level | Satellite image observation and analysis available |
| | | Land structure | The proportion of each type of land | Data comparison before and after planning |
| | | Spatial structure | | |
| | Real estate value | Residential sales price | | |
| | | Rent | | |
| | Job | Number of jobs created | | |
| | Tourism consumption | Site rental fee | | |
| | | Ticket income | | |
| | | Sales revenue at the scenic spot | | |
| Social performance | Public facility | Education | Whether the construction of public facilities is in accordance with the plan, construction level and effectiveness | Government statistics, work report and interview |
| | | Medical | | |
| | | Culture | | |
| | | Sport | | |
| | | Pension | | |
| | Municipal infrastructure | Traffic | Whether the construction of public facilities is in accordance with the plan, construction level and effectiveness | |
| | | Water supply | | |
| | | Drainage | | |
| | | Electric power | | |
| | | Telecommunications | | |
| | Related policy | Public participation | Awareness of government-related policies | |
| | | Satisfaction | Residents' satisfaction with government-related policies | |
| | Safe society | Flood | | |
| | | Fire | | |
| | | Earthquake | | |
| | Culture | Ancient tree protection | | Site research and interviews |
| | | Historical and cultural heritage protection | Whether the construction caused a breach of the ring; whether there is protection and repair work; whether it is integrated with tourism development | |
| | | Cultural activity | What specific cultural practices and cultural activities are carried out | |
| Ecological performance | Ecosystem | Urban air particulate reduction | | |
| | | Water source protection | Whether to protect according to the planning expectations, and the protection status quo | Government statistics and interviews |
| | | Agricultural and forestry protection | | |
| | | Water quality | Main rivers and lakes | Data comparison |
| | | Rainwater and sewage construction | Related construction situation | Main landscape node site survey |
| | | Rainwater management | | |
| | | Enterprise pollution transformation | Industrial and aquaculture pollution treatment situation | Site research and interviews |
| | | Environmentally friendly materials use | Whether it is implemented according to design | Main landscape node site survey |
| | Landscape environment | Green space system | Planning and construction | Government work report and interview |
| | | Green coverage | | |
| | | Per capita greening rate | Implementation rate | Government statistics |
| | | Main landscape node construction style | Construction status of the consistency with the planning | Photo comparison and interviews to understand the construction situation |
| | | Street environment | Health status, interface appearance, vehicle order | |
| | | Community environment | Sanitary facilities situation | |
| | | Landscape satisfaction | Landscape view | Research |

Notes: ▓ Village and town planning implementation assessment other content factors; ▓ The Beautiful Rural Construction evaluation index factors; ▓ Screening of American LP evaluation index factors; ▓ Zhejiang Province Small Towns Environmental Comprehensive Improvement Action Township Assessment Score factors.

The new indicator system adopts the US "landscape performance" evaluation system as the first-level indicator framework, of which the theoretical framework is also based on the three elements of sustainable development (environment, economy, and society). The determination for the secondary indicators and tertiary indicators is based on the verification of practices of village and town construction guided by "Ecological Priority and All-Area Integrated Development". This new system, through a comparative index study, is an implementation of the traditional government-led evaluation model.

The overall evaluation system is set for the comprehensive performance evaluation on the mesoscale, or intermediate scale level. As far as the village and town construction level is concerned, the traditional "landscape performance" indicators are too micro. The overall urban planning implementation evaluation indicators are relatively macroscopic, while the small-town acceptance criteria are relatively specific. In our opinion, the newly composed evaluation criteria for village and town construction should be a combined use of micro and macro indicators. It is a comprehensive study based on qualitative and quantitative research.

In the following case studies, the full Construction Performance Evaluation processing routine was applied to assess Shilin Village (Case 1), and Dalan Town (Case 5). The other four case study results were also summarized.

## 4. Case Study Results

### 4.1. Case Study 1: Landscape Implementation for Shilin Village

The landscape implementation goal is to solve the problems of insufficient space in the scenic area of the persimmon village, the tourist reception capacity, the lower level of reception and the protection of the ancient villages [20].

Through on-site investigation, field study, and questionnaire surveys, the AHP, and FCE methods were used to compare and analyze the construction before and after planning (as shown in Table 3.)

**Table 3.** Shilin Village Construction Performance Evaluation Case Study.

| Post-Construction Evaluation Indicator System for Coordinated Development of Rural Communities & Small-Towns Based on "Ecological Priority and All-Area Integrated Development" | | | | | Landscape Construction Planning (Shilin Village–2014) | | | Analytic Hierarchy Process and Fuzzy Comprehensive Evaluation | | | | | | |
|---|---|---|---|---|---|---|---|---|---|---|---|---|---|---|
| | | | | | | | | | Classification | | | | | |
| First Level | Second Level | Third Level | Remark | Source | Pre-Planning Status | Planning Objectives | Verifying Status | Weights | Best (2) | Better (1) | General (0) | Bad (−1) | Worse (−2) | Score |
| Economic performance | Industrial Economy | Visitors volume | | Government statistics | Total number of tourists in 2015 was 500,000, visitors per day were 0.15 million/day. | The planned average daily tourist volume is 0.23 million | ●After the planning, the number of tourists has increased | 0.053 | 6 | 2 | 1 | 0 | 0 | 14 / 15.0 |
| | | Parking revenue | Mainly for tourists parking | | ○ | ● | ●Increased parking lot revenue after planning | 0.053 | 7 | 2 | 0 | 0 | 0 | 16 |
| | Real estate value | Residential sales price | | | ○ | ○ | ● | 0.053 | 5 | 3 | 1 | 0 | 0 | 13 / 12.0 |
| | | House rental fee | | | ●Accommodation was two-star level | ● | ●After the planning, the price of accommodation is raised. | 0.053 | 3 | 5 | 1 | 0 | 0 | 11 |
| | Job | Number of jobs created | | | ● The service industry level was low | ●Increasing tourism services; Improving the services quality | ●Tourism service projects have increased and the number of jobs has increased | 0.053 | 4 | 5 | 0 | 0 | 0 | 13 13.0 / 12.88 |
| | Tourism consumption | Site rental fee | | | ○ | ● | ●Increased rental revenue after planning | 0.053 | 3 | 5 | 1 | 0 | 0 | 11 |
| | | Ticket revenue | | | ●Ticket revenue is relatively stable, high income in summer and autumn season | ○ | ●Increased whole year ticket sales after planning | 0.053 | 6 | 1 | 2 | 0 | 0 | 13 / 12.0 |
| | | Internal consumption within scenic areas | | | ●23 Farmhouses, 1980 dining places, 91 beds homestay of general consumption level | Plan to increase 560 beds, improvement of dining grade | ●The number of homestays has increased, and the level of dining has not improved much. | 0.053 | 4 | 4 | 1 | 0 | 0 | 12 |
| Social performance | Cultural style | Ancient tree protection | | Site research and interviews | ○ | 16 ancient trees protection planning | Key ancient tree brand protection | 0.053 | 4 | 5 | 0 | 0 | 0 | 13 |
| | | Historical and cultural heritage protection | Protection and repair work situation; tourism development integration situation | | Historical and cultural buildings were generally protected; the overall construction quality was poor | To repair and protect historical and cultural buildings, to improve the surrounding environment | Repairing and improvement effect is obvious, the surrounding environment has improved | 0.053 | 5 | 2 | 2 | 0 | 0 | 12 12.0 12.00 |
| | | Cultural activity | Types of specific cultural practices and activities | | Existing "Shen Clan" Sacrifice Culture, the Persimmon Festival | Diversifying cultural and folk activities | Implementation of the red cultural activities | 0.053 | 5 | 1 | 3 | 0 | 0 | 11 |

**Table 3.** *Cont.*

| Post-Construction Evaluation Indicator System for Coordinated Development of Rural Communities & Small-Towns Based on "Ecological Priority and All-Area Integrated Development" | | | | | Landscape Construction Planning (Shilin Village–2014) | | | Analytic Hierarchy Process and Fuzzy Comprehensive Evaluation | | | | | | | |
|---|---|---|---|---|---|---|---|---|---|---|---|---|---|---|---|
| | | | | | | | | | Classification | | | | | | |
| First Level | Second Level | Third Level | Remark | Source | Pre-Planning Status | Planning Objectives | Verifying Status | Weights | Best (2) | Better (1) | General (0) | Bad (−1) | Worse (−2) | Score | |
| Ecological performance | Ecosystem | Water quality | Main rivers and lakes | Data comparison | ●Most of the water resources were type II water | Dredging the river to improve the quality of the water environment | ●Water quality has improved | 0.053 | 5 | 3 | 1 | 0 | 0 | 13 | |
| | | Rainwater and sewage construction | Related construction situation | Main landscape node site survey | Frequent flood disasters | Dredging the river channel to improve flood discharge capacity; greening on both sides of the mountain river channel to enhance water storage capacity; widening the river section to consolidate and strengthen flood control capacity | The implementation of rainwater management construction is better, no stormwater disaster occurs during the year. | 0.053 | 4 | 4 | 1 | 0 | 0 | 12 | 12.0 |
| | | Environmentally friendly materials usage | Whether it is implemented by design | Main landscape node site survey | Constructed with mostly low environmentally friendly materials | To use native stone for building walls, advocating native plants usage | Partially remodeled buildings use native stone, and more native plants are used. | 0.053 | 5 | 2 | 1 | 1 | 0 | 11 | |
| | Landscape environment | Green coverage | | | Village green coverage was high | Greening along the road line to enrich the plant landscape | The planning area is good in richness and variety. | 0.053 | 4 | 2 | 2 | 1 | 0 | 9 | 12.13 |
| | | Main landscape node construction style | Construction status, consistency with planning | Photo comparison and interviews to verify the construction situation | Node landscape was poor | Transforming and upgrading important landscape nodes, featuring the cultural context | The implementation is basically consistent with the plan, and the effect is better. | 0.053 | 2 | 6 | 1 | 0 | 0 | 10 | |
| | | Street environment | Cleanness status, Street scape, parking order | | Monotonous view, rough street furniture, parking chaos | Improving the street environment to create a clean and tidy street, to build a new garage for village vehicles | The street environment is upgraded to a higher level, the street facilities have been upgraded, the new garage has a high utilization, village parking more orderly. | 0.053 | 7 | 2 | 0 | 0 | 0 | 16 | 12.2 |
| | | Community environment | Sanitary facilities situation, and landscape view | | Traditional building structures were aging, the public facilities were not perfect | Demolition and repair of certain buildings | The overall community environment has been improved | 0.053 | 5 | 4 | 0 | 0 | 0 | 14 | |
| | | Landscape satisfaction | | Research | ○ | High satisfaction | ●Higher satisfaction | 0.053 | 4 | 4 | 1 | 0 | 0 | 12 | |
| | | | | | | | | 1 | | | | | | | 12.421 |

Notes: ● Data to be collected, ○ Missing data; ▮ Village and town planning implementation assessment other content factors; ▮ The Beautiful Rural Construction evaluation index factors; ▮ Screening of American LP evaluation index factors; ▮ Zhejiang Province Small Towns Environmental Comprehensive Improvement Action Township Assessment Score factors.

In the evaluation process, the equal weight assignment evaluation method was adopted for each indicator input. The performance scores of the comprehensive improvement plan along the village were all greater than 0: economic performance score was 12.88, social performance score was 12.00, and ecological performance score was 12.13.

*4.2. Case Study 2: Small Town Remediation Planning Project—Luting Township Market Town*

The remediation project was to build the "coordination of tourism development and township construction" based on the existing scenic belt, creating a new brand for rural tourism. The key construction part is village appearance implementation on both sides of the road along the Heliang Road development axis [21].

In the evaluation process, the equal weight assignment evaluation method was adopted for each indicator input. The performance scores of the comprehensive improvement plan of Luting Township were all greater than 0: Economic performance score was 3.50; social performance score was 8.73, ecological performance score was 7.91, and comprehensive score was 6.96.

*4.3. Case Study 3: Village Planning Project—Hengkantou Village*

The plan of Hengkantou Village achieved four goals: (1) To restore the original spatial structure of the damaged villages. (2) Improvement of tourism reception infrastructure. (3) To prevail in development competition on the basis of featured tourism. (4) To set a leading example in the realization of "new industrialization and urbanization linkage" [22].

In the evaluation process, the equal weight assignment evaluation method was adopted for each indicator input. The performance scores of Hengkantou beautiful rural rectification plan were all greater than 0: economic performance score was 11.07, social performance score was 13.778, ecological performance score was 13.78 and composite score was 12.59.

*4.4. Case Study 4: Tourism Planning Project—Along the Ring of Siming Lake*

The plan was to build a robot theme park along the Siming Lake, with implementation of a leisure and holiday resort service system around the lake. The whole functional group was to integrate with the overall service system of leisure tourism in Liangnong Township and merge with the overall development plan of Siming Mountainous area [23].

In the evaluation process, the equal weight assignment evaluation method was adopted for each indicator input. The performance scores of the tourism planning projects along the Siming Lake were all greater than 0: economic performance score was 7.46, social performance score was 9.50, ecological performance score was 10.80, and the final combined score was 9.46.

*4.5. Case Study 5: Concept Planning of Tourism Development in Dalan Town (2015)*

Over the past decade, Dalan Town had undergone rapid all-area tourism development. As the various tourism supporting infrastructure is continuously improved, accommodation and catering for tourism entertainment and leisure activities, and public toilet services have all been implemented [24].

In the evaluation of the tourism development plan of Dalan Town, each indicator sub-item adopts the equal-weight assignment evaluation method, and the performance scores (transportation infrastructure and vacation facilities) were greater than 0: economic performance score was 18.67, social performance was 13.00, ecological performance score was 19.00, and the overall performance score was 16.89 (as shown in Table 4.)

**Table 4.** Construction performance evaluation case study featuring parking facilities sharing and tourism and leisure facilities in Dalan Town.

| Post-Construction Evaluation Indicator System for Coordinated Development of Rural Communities and Small Towns Based on "Ecological Priority and All-Area Integrated Development" (Transportation Infrastructure and Leisure Facilities) | | | | | Concept Planning of Tourism Development in Dalan Town (2015) | | Analytic Hierarchy Process and Fuzzy Comprehensive Evaluation | | | | | | |
|---|---|---|---|---|---|---|---|---|---|---|---|---|---|
| | | | | | | | | Classification | | | | | |
| First Level | Second Level | Third Level | Source | Remark | Analysis of the Status Quo before Planning and Construction | Verifying after Planning and Construction | Weights | Best (2) | Better (1) | General (0) | Bad (−1) | Worse (−2) | Score |
| Economic performance | Industry | Visitors volume | Statistical report | | Catered more than 6.7 million visitors in 2014 | Visitors increased | 0.10 | 3 | 4 | 2 | 0 | 0 | 10 |
| | | Resident employment type | Questionnaire | General type of employment | ● | The number of employed people in tertiary employment has increased year by year | 0.10 | 3 | 3 | 3 | 0 | 0 | 9  7.67 |
| | | Parking revenue | | Mostly for tourists parking | ○ | Revenue largely increased in the peak season | 0.10 | 0 | 5 | 3 | 1 | 0 | 4 |
| | Real estate value | House rental fee | | | ● | Rental fee significantly Increased | 0.10 | 1 | 2 | 4 | 2 | 0 | 2  2.00   18.67 |
| | Jobs | Number of jobs created | | | ○ | Tourism packages such as boutique hotels and hotels have greatly boosted employment in the tourism industry | 0.10 | 2 | 3 | 4 | 0 | 0 | 7  7.00 |
| | Tourism consumption | Site rental fee | | | ● | ● | 0.10 | 0 | 3 | 5 | 1 | 0 | 2  2.00 |
| Social performance | Municipal infrastructure | Traffic | | Facilities Construction accordance with the plan, and construction effectiveness | One existing gas station; main road frame consist of two road lines, linking various tourist attractions and administrative villages | One tourist distribution center, three bicycle stations, 12 scenic spots VR showcase pavilions are added; over 10 bike riding trails, car driving lines, and walking trails are planted. | 0.10 | 5 | 3 | 1 | 0 | 0 | 13  13.00  13.00 |
| Ecological performance | Ecosystem | Environmental friendly materials usage | Main landscape node site survey | Whether it is implemented according to design | Villages roads were mainly old stone pavements | Using local materials such as stones and permeable grass plants for landscape implementation | 0.10 | 2 | 4 | 3 | 0 | 0 | 8  8.00 |
| | Landscape environment | Main landscape node construction style | Photo comparison and interviews to verify the construction situation | Construction status, consistency with planning | The landscape features along the main line were good | Key scenery and vista landscape nodes are further more improved | 0.10 | 4 | 3 | 2 | 0 | 0 | 11  11.00   19.00 |
| | | Landscape satisfaction | Research | | General | Better | 0.10 | 3 | 5 | 1 | 0 | 0 | 11 |
| | | | | | | | 1 | | | | | | 16.89 |

Notes: ● Data to be collected, ○ Missing data; <span style="background:#9DC3E6">▮</span> Village and town planning implementation assessment other content factors; <span style="background:#FFE699">▮</span> The Beautiful Rural Construction evaluation index factors; <span style="background:#A9D18E">▮</span> Screening of American LP evaluation index factors; <span style="background:#F4B183">▮</span> Zhejiang Province Small Towns Environmental Comprehensive Improvement Action Township Assessment Score factors. Source: "Dalan Town Tourism Development Concept Plan" prepared by Shanghai Xilian Urban Planning Architectural Design Co., Ltd.; "Dalan Town Master Plan (2011–2030) Partial Revision" (2016) prepared by Zhejiang Jianyuan Architectural Planning and Design Institute.

*4.6. Case Study 6: Lubu Town's Overall Tourism Development Plan (2017)*

Lubu Town of Yuyao City is a traditional industrial town with a focus on industrial development. In recent years, the Lubu Town Government has changed the original industrial-centric development policy to promote economic transformation to increase the excavation of tourism resources and address the increasing pressure on environmental protection. In this study, parking facilities, and concentration of tourism and leisure service facilities were selected to indicate infrastructure performance [25].

In the evaluation of the overall tourism development plan of Lubu Town, each indicator sub-item adopts the equal-weight assignment evaluation method. The performance scores (transportation infrastructure and vacation facilities) were all greater than 0: economic performance score was 8.00, social performance was 6.00, ecological performance score was 9.50, and the overall performance score was 7.83.

*4.7. Comparative Study and Comprehensive Analysis for Key Findings*

4.7.1. Findings for Ecological Priority

First, a comprehensive evaluation for the actual construction performance of Case study 1–4 was conducted. Then the newly composed evaluation index system factor based on US "landscape performance" evaluation was assigned, scored and compared. Finally, the comparative evaluation method was used to conduct a comprehensive comparative analysis (as shown in Tables 5 and 6).

**Table 5.** First-level performance indicator statistics towards ecological priority orientation.

|  | Luting Town | Shilin Village | Along the Ring of Siming Lake | Hengkantou Village |
|---|---|---|---|---|
| Economic performance | 3.50 | 12.86 | 7.46 | 11.07 |
| Social performance | 8.73 | 12.00 | 9.50 | 13.78 |
| Ecological performance | 7.91 | 12.13 | 10.80 | 13.78 |
| Comprehensive performance | 6.96 | 12.42 | 9.45 | 12.59 |

Through quantitative comparison of the construction performance for different types of projects, the relatively high and low scores of economic, social, and ecological performances were observed. Comprehensive performance of the project can be obtained separately.

**Table 6.** Secondary level performance indicator statistics towards ecological priority orientation.

| Evaluation Index System After Comprehensive Planning | | Study Sample Typical Case Index Selection | | | |
|---|---|---|---|---|---|
| | | Project Type | | | |
| Level 1 | Level 2 | Village Planning (Hengkantou Village) In 2013 | Landscape Planning (Shilin Village) in 2014 | Tourism Planning (Along the Ring of Siming Lake) in 2016 | Comprehensive Improvement Project of Small Towns (Luting Town) In 2017 |
| Economic performance | Industrial economy | 12.8 | 15 | 8.5 | 1.5 |
| | Social development | 9 | / | / | / |
| | Space usage | 10.3 | / | 10 | 3 |
| | Real estate value | 11 | 12 | 7.5 | 4 |
| | Jobs | 12 | 13 | 10 | 6 |
| | Tourism consumption | 10 | 12 | 6.7 | 5 |
| Social performance | Public service facilities | 13.8 | / | 8 | 8.8 |
| | Municipal infrastructure | 13.8 | / | 12 | 8 |
| | Relevant policy | 13 | / | 7 | 5 |
| | The social security | 10.7 | / | 11 | 4 |
| | Cultural landscape | 14.3 | 12 | 11.3 | 10 |
| Ecological performance | The ecological environment | 10.4 | 12 | 11.3 | 8.4 |
| | Landscape environment | 12.8 | 12.2 | 9 | 7 |

Through horizontal data comparison, the final result shows that the overall performance level is basically consistent with the ranking of ecological performance scores.

It can be seen from Figure 10 that the project's overall performance is consistent with the ranking of ecological and social performance. To a certain extent, it reflects the importance of ecological priority orientation in the construction process of villages and towns in Siming Mountainous area of Zhejiang.

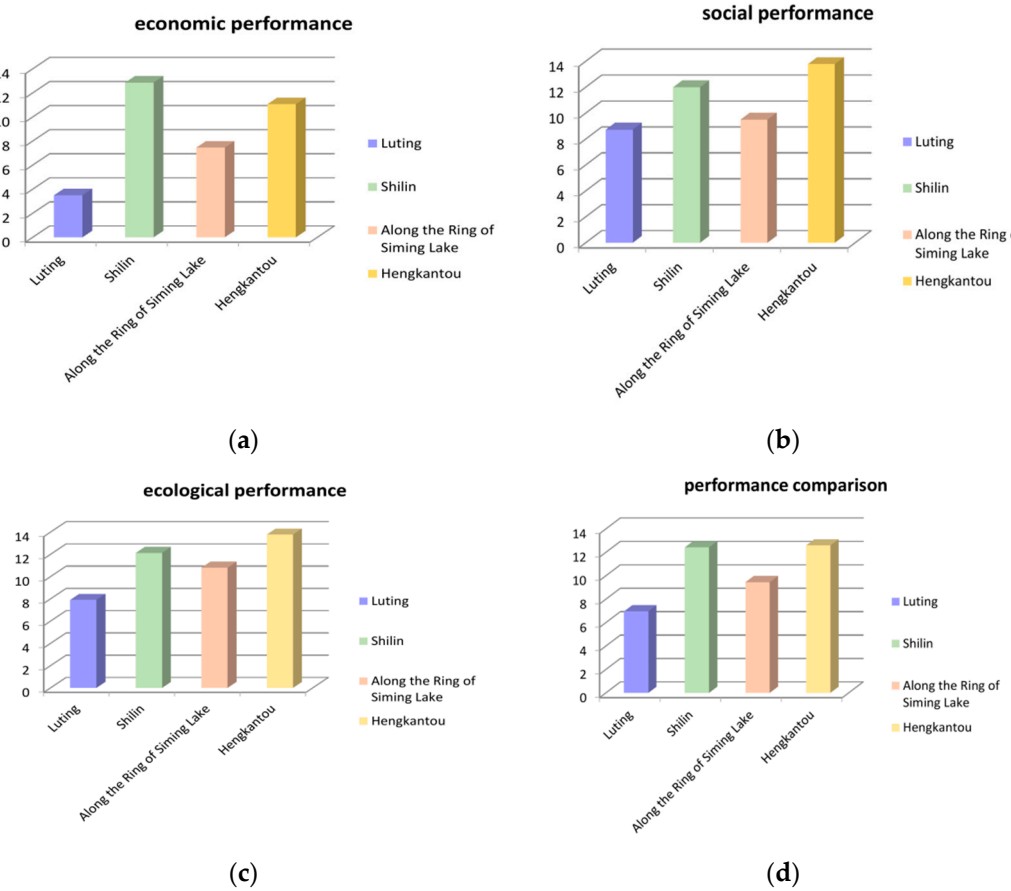

**Figure 10.** Project performance comparison chart towards ecological priority orientation. (**a**) Comparison of economic performance. Shilin > Hengkantou > Along the Ring of Siming Lake > Luting, (**b**) Comparison of social performance. Hengkantou > Shilin > Along the Ring of Siming Lake > Luting, (**c**) Comparison of ecological performance. Hengkantou > Shilin > Along the Ring of Siming Lake > Luting. (**d**) Comprehensive performance comparison. Hengkantou > Shilin> Along the Ring of Siming Lake > Luting.

Meanwhile, the pre-post analysis shows that the concept of "Ecological Priority and All-Area Integrated Development" is already blended into the planning and construction implementation process. After three to five years of construction, the performance scores were all greater than zero, indicating that the construction achievements are more integrated to a certain extent. Under the concept guidance of the all-area coordinating of villages and towns, each construction project can give it full play to maximize the regional sharing of resources and public facilities.

It is worth noting that this kind of comparison method has limitations, and the amount of data collected in the historical period is relatively insufficient. More research is needed to strengthen future verification.

4.7.2. Findings for All-Area Integrated Development (Analysis of Parking Facilities and Leisure Facilities Indicator)

The last two sample cases are selected from the contents of the town-wide tourism development plans. In this study the demonstration of construction performance is indicated through the aspects of parking facilities and related service facility-sharing. The two towns are all in the Siming Mountainous area. The tourism development boom of Dalan Town happened earlier, resulting in a better overall performance score of transportation infrastructure and leisure events facilities. The Lubu Town, meanwhile, used to be a traditional industrial town, which has gradually transformed in recent years.

The concept of "All-Area Integrated Development" has been leading the practice of village and town construction in the eastern Zhejiang region for a long time. The all-area tourism construction is the most direct practice based on this development concept. A comparative study of the evaluation (the parking facilities and leisure service facilities construction) of the two towns found that the performance scores and other aspects are all positive, highlighting that the towns' development is on the right path. The performance of Dalan Town (in the sharing of parking facilities and the supporting leisure events facilities) is significantly better than that of Lubu Town (as shown in Figure 11). To a certain extent, it also indicated that the effects of All-Area Integrated Development can only be obtained in the long run, since it needs a comparatively whole area integration.

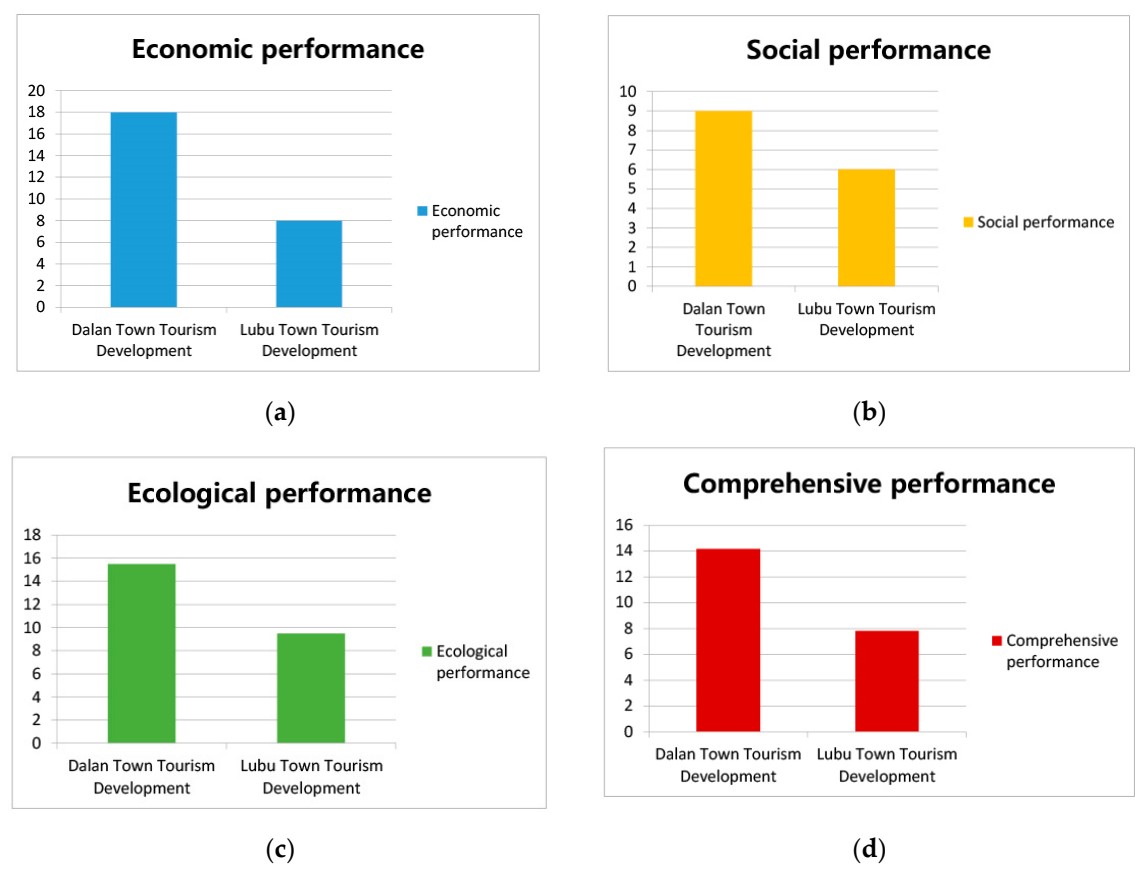

**Figure 11.** Project performance comparison chart towards All-Area Integrated Development. (**a**) Comparison of economic performance. Dalan>Lubu. (**b**) Comparison of social performance. Dalan > Lubu. (**c**) Comparison of ecological performance. Dalan > Lubu. (**d**) Comprehensive performance comparison. Dalan > Lubu.

This study is intended to demonstrate the use of the new comprehensive evaluation system. For in-depth understanding of the eastern Zhejiang region, more data and further investigation is needed.

## 5. Conclusions

In this research, the proposed comprehensive post-construction evaluation system based on "Ecological Priority and All-Area Integrated Development" is highlighted as being different from the traditional rural planning and development evaluation systems in China. By introducing the US "landscape performance" concept to existing systems, the new proposed system is more suitable for development evaluation of rural communities and small towns.

This new system is built on comprehensive performance evaluation at the mesoscale level. It is a combined use of micro and macro indicators. Through a combination of qualitative and quantitative methods, the system cites the first-level evaluation framework from the US "landscape performance" model. The overall theoretical framework is based on the three elements of sustainable development—environment, economy, and society.

We adopted this new system for six case studies located in the Siming Mountain area of East Zhejiang, China. The results indicate that the Dalan Town Tourism Development and the Shilin village landscape belt construction of Dalan showed higher performance scores. This result correlates with the unique landscape heritage advantages and livable rural residence community compatibility of the areas in our case studies. The ecological harmonious characteristics of the small-town area greatly benefit sustainable interaction of government, developers, and local community for future development priority of Dalan in the Siming Mountain region. As for the case study of Lubu Tourism Planning and Development Construction, our study found that the town should pay more attention to transforming the existing low-level industry. It should also take smart moves to integrate resources based on the "Coordinated Development" idea.

In this research, "Ecological Priority" refers to both natural ecology and human ecology. "Coordinated Development of Rural Communities and Small Towns" is implied at the spatial resources and rural socio-economic development level, which significantly enhances the highly effective resource-saving development mode. In this process, the development-led intervention based on landscape performance evaluation is conducive to the development of the overall model, which is more scientific and comprehensive, with anticipated performance outcomes. The new evaluation system can be adopted by state government planning departments, tourism agencies, real estate developers, etc. to allow for more informed, environment protection and integrated development decision-making. It can also be a great resource for future application in southeast Asian countries that have similar development patterns and climatic conditions, in a bid to conduct comprehensive evaluation of the vast area of rural communities and small towns.

Although this search uses certain construction evaluation data from 2012 to 2018, it lacks continuous tracking of many key level performance data for non-quantitative evaluation. The swift changes in rural development policies over the past decade have also had a certain impact on the evaluation results.

**Author Contributions:** Conceptualization, T.Z. and Y.Z.; Methodology, Y.Z.; Formal analysis, T.Z.; Investigation, T.Z. and Y.Z.; Writing—original draft preparation, T.Z.; Writing—review and editing, M.-H.L.; Supervision, L.M.

**Funding:** This research received no external funding.

**Acknowledgments:** This research is a collective effort of consistent planning and design, and consultant works over 10 years, undertaken by the authors' planning and design team in the cases study area. In an oral presentation at CELA 2018, the authors' team shared certain original research ideas of this paper with the audience to engage in discussions with scholars in the field [26]. Our postgraduates Binghua Li, Miao Yang, You Liu, Jing Ning, Yuanyuan Gao, Xiaotian Zhao had devoted their continuous efforts to field study and on-site case studies. The questionnaires, and policy and planning information was mainly obtained from local administration of the Yuyao Planning Bureau, and from the governments of Dalan, Liangnong, Lubu, Luting and related villages.

**Conflicts of Interest:** The authors declare no conflict of interest.

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
