# Peer review of "Landscape Performance for Coordinated Development of Rural Communities & Small-Towns Based on “Ecological Priority and All-Area Integrated Development”: Six Case Studies in East China’s Zhejiang Province"

_sustainability, doi:10.3390/su11154096_

Round 1
Reviewer 1 Report
Summary Comments:
Thank you for the submission and opportunity to review. I do appreciate authors’ effort to integrate various evaluation approaches to better understand rural communities and small towns. The research presented seems interesting but needs additional work to get the publication quality in my view. Following are some thoughts for authors and/or journal editorial team to consider.
Title: Title implies single case study whereas the paper covers up to 6 case studies. I believe title should be revised for clarity and language.
Abstract: I find the abstract informative but little too broad. I would encourage author(s) to clarify purpose statement and shed some lights on all sections of a conventional research framework namely Introduction/Problem Statement, Background/Literature, Methodology, Analysis/Findings/Results and Conclusion/Discussion. Especially the limited focus on methods as well as the study sites starting in the abstract seem to be a continuing limitation throughout the paper.
Introduction/Background/Literature:
- I would urge authors to explain in brief words what is the problem statement for this research while trying to respond to “so what” question? Why do we need this research?
- I appreciate part of the coverage in background and literature. That being said this section doesn’t seem to fully cover references especially in landscape performance studies. I would urge authors to include important/critical literature in this area. For example:
1- LAF Studies and resources should be properly cited and referenced throughout the document as well as in the final section. Perhaps LAF Guideline and toolkits can be referenced.
2- Two China based journals have special issues and/or significant coverage on landscape performance studies. If they are not covered they should be.
a. Journal - (May, 2016). Landscape Architecture Frontiers – LAF, 4(2)
b. Journal - (February, 2015). Landscape Architecture Journal hosted by Beijing Forestry University, China (a bimonthly peer-reviewed academic journal). Special Issue
3- I also believe that authors have a presentation on the topic at CELA so that should be acknowledged.
- Please consider inserting purpose statement and research objectives and questions clearly as a paragraph or sub section.
Methods:
- Major point of concern for the paper to me is Methodology. I believe it is very limited, even though bits and pieces of procedures seem to be covered in some other sub-sections throughout the paper. I would urge author(s) to have a consolidated, complete and clearly articulated methodology section that fully covers; research paradigm, research design, study location and population, data collection, analysis methods. The value of the paper in my view can mostly be articulated with replicable and generalizable procedures and methods and the paper at this stage in my view requires this attention.
Analysis & Results:
- I appreciate variety of methods explored by the authors. However, the results for case studies are mostly given in the form of tables and very repetitive paragraphs in terms of language. I would urge authors to present the work through insightful analysis and synthesis of the findings rather than just providing tables, charts and diagrams with limited text.
- Another major point for me that must be addressed is that I am truly not clear why four case studies and the two case studies analyzed separately? If the research suggests that it follows completed/constructed projects as “landscape performance” approach suggests why they are not combined? If they are different type of case studies and require different evaluation systems why they are covered in the same paper? Authors owe reader critical explanation of the analysis procedures.
Discussion & Conclusion:
- Conclusion seem to have basic essentials but it is very very brief and very little discussion is presented. Once again I would urge authors to explain why the topic does matter and relevant as part of the body of Sustainability Journal? Please try to answer what does this all mean? What are lessons to draw for the reader? Perhaps any future directions?
Language & Formatting:
- Paper requires significant editing for clarity, precision, grammar, punctuation, and formatting.
- Some phrases, and terms may also have to be checked for language and/or defined for layman.
- Overall - Organization of the paper is confusing and not clear where introduction/, literature, methods, analysis & results, discussion & conclusions starts and ends. Please consider revising some of your subtitles throughout the document for simplicity and clarity.
Line-by-line (sample comments):
Following are only couple of sample line-by-line issues that needs further clarity by authors. Authors can look at these as sample issues and try address them throughout the paper.
Line - 25 What specific objectives?
Line- 34 & 40 Are these titles or sentence? Please revise them as titles.
Line - 38-39 Check language
Line - 49-50 Check language
Line - Starting with Line 80 - Please indicate the research methods/paradigm using appropriate references. Indicate relevant literature, whether the work is qualitative or quantitative? Methodology needs improvements.
Line – 116 Ones?
Line – starting with 176 - In my view methodology should be discussed under methodology and some of the subject matter (177-187) here more appropriate for 3.1. Please also insert LAF citations/references/
Line – 211 Table can be formatted better to make it easy to read.
Line – 253- Paragraph formatting issues
Line 289 – 298 Formatting as well as scores can be rounded to one digit after period to simplify the results.
--
Due to limited time provided by the journal, as well as added extra pressure to turn this review in earlier than the agreed time by editorial team I will only be able to provide above comments at this time.
In my view, paper has potential but it needs major revisions. Thanks to authors for this interesting work, and looking forward for revised and improved version, if it was accepted by the journal.
Sincerely,
Author Response
Thank you so much for your kind review and comments! Please see to the uploaded files of cover letter for revise and revised manuscript for the detailed reply.

Reviewer 2 Report
This paper reviews and integrates the current sustainability assessment systems for community development in China with one of the counterparts in the United States, i.e., Landscape Architecture Foundation’s metrics and evaluation system, and applies this integrated new system to evaluate six cases in Zhejiang Province in China. The paper contains more than 10 years of field work and analysis conducted by the research team. Research findings would be helpful to guide professionals who practice in this area.
I would suggest adding more discussions on the potential limitations of the proposed new system and how the research findings can be generalized to use in other regions in China. Also, why choosing these six cases? Would be helpful to talk more about the rationale. The paper mentions that these cases represent “6 typical types of village and town construction projects”. I suggest adding a paragraph to introduce these terms… the “village planning case, garden construction planning case, tourism planning case, small-town environment comprehensive remediation case.”
Author Response

(The authors gave the same response as above.)

Reviewer 3 Report
This paper demonstrates an approach for creating an integral evaluating system for rural communities in China. This work is interesting and of relevance to the current advocacy of evidence-based design. Evaluating the communities that developed under the recently published guidelines can not only test these guidelines’ effectiveness, but also inform future decision making, and therefore contribute to the success of sustainable rural communities. Below are some suggestions for further improvement.
The paper can elaborate on why there is a need to create an evaluation system to assess all types of rural communities, e.g. allow cross-sectional comparison between different projects. In addition, it might be better to further clarify why landscape performance is selected as the base to develop this new system on.
In addition, more details are needed to help readers understand the exact method. For example, how are the various metrics selected and how are the score and weight systems combined?
Lastly, in the discussion/conclusion section, it would be great if the authors can discuss the wider international influence and relevance of this evaluation system.
Author Response

(The authors gave the same response as above.)

Reviewer 4 Report
It is very hard to judge what is the purpose of the article: is it a description of some formal regulations?
The introduction is too generalized. Non-Chinese reader may have no idea how to refer to some points, for example, the article lacks some basic desription of provided chinese regulations. Without this knowledge it's impossible to get any insights.
Also, it's unclear what is the research aim and research method. The table no. 1 puts some values but they are unsatisfactory. The authors should provide some closer info what has been their "field research", "field investigation" and so on. Then, what is the purpose of figure no. 1? The drawings are at very general level, showing some basic information. I would be really interested, how authors have used these informations? Then, I'd say the given supply chain is well-known since the very first settlements, so what is so important in it now? Frankly, nowadays I'd expect some deeper spatial analysis, using some modern methods (showing real time access, some weight of freights between spaces or something like this). That's true, there are some numbers but I cannot figure out how these values have been calculated, especially with such precision. I understand, the tables no. 3 and 4 give us some numbers, but what do they mean? Aren't the values taken arbitrary?
And regarding these numbers: I would love to know what is the ralation between "planned average daily tourist volume of 0.23 milion" and "plan to increase 560 beds"?
Finally, I wonder why there are 3 criteria for the landscape evaluation (economic, social and ecology performance) but there is no mention about any visual element, such as composition, heritage values etc. I wonder what is the authors comprehension of the landscape? Is it a value that can be developed and modified? Or is it something that should be preserved? The table no. 2 shows a list of land-use objectives and development. How these values refer to landscapes?
Overall, the article seems rather to be a checklist for planners. It looks like the authors wanted to show what had been their background for selecting the two regions (Shilin and Hengkantou) for providing some financial support for them.
Author Response

(The authors gave the same response as above.)

Reviewer 5 Report
The manuscript deals with an interesting topic focused on a new comprehensive evaluation system rooted in the US ‘landscape performance’ concept, which was applied in the Chinese context. The manuscript is difficult to read and its structure is unclear. The aims are unclear and should be clearly stated in both ‘Abstract’ and ‘Introduction’. A detailed analysis of the scientific literature is missing and it is unclear if the study is rooted in scientific basis and considers proper references and research gaps. ‘Section 2’ and the discussion of the findings are missing. The authors should clearly discuss about the relevance of their study compared to the international scientific panorama and point out the research gaps they filled in with their study. The manuscript shows serious pitfalls that should be addressed by the authors and in my opinion it is not ready for publication.
Author Response

(The authors gave the same response as above.)

Round 2
Reviewer 1 Report
Summary Comments:
Thank you for the revised submission and opportunity to review this second time. I do appreciate authors’ effort to improve the paper to a quality anticipated. Authors’ seem to respond most of my comments. Following are some minor thoughts for authors and/or journal editorial team to consider. Language and formatting still seem to have issues. I would urge authors to find a stronger professional editor.
Title: Improved.
Abstract: Improved. It may be good idea to introduce two objectives, integrated only in the methodology, in the abstract and/or introduction as well. Parts of these objectives can be inserted in to the abstract for consistency.
Introduction/Background/Literature: Improved. It looks much stronger with stronger literature coverage.
- Mentioned in review.1 - I believe that authors have a presentation on the topic at CELA so that should be included in the references.
- Mentioned in review.1 – I believe not acknowledging the following journal issue in references - (May 2016). Landscape Architecture Frontiers – LAF, 4(2) is a missed opportunity unless the following paper comes from this edition. Is it from this journal or some other? If it is please check reference details for accuracy.
[18] BoYang; Shujuan Li; Chris Binder. A research frontier in landscape architecture: landscape performance and assessment of social benefits[J]. Landscape Research,2016,41(3).
Mentioned in review.1 - I believe not acknowledging following journal issue in references - (February, 2015). Landscape Architecture Journal hosted by Beijing Forestry University, China (a bimonthly peer-reviewed academic journal). Special Issue, is a missed opportunity unless the following reference comes from this edition. I believe proper citation for the below reference may also need to include “Landscape Architecture Journal, Special Issue, p70-86.”
[20]Ozdil, T; R., & Stewart, D.M. (2015). Assessing economic performance of landscape architecture projects: Lessons learned from Texas case studies. Landscape Architecture. Available online: http://www.la-bly.com/news/show_news.aspx?news_id=208
Methods: Improved - Although it is much better I find methodology and procedures a bit obscured and unclear in organization making it problematic for the reader to follow.
- Section.3 Methodology sub titles does not make sense. Why do authors have “3.3. Introduction to Research Methods” after section 3.1 & 3.2? (see line 200). Titles and/or organization is confusing in this section. Authors must check and consider revising titles and sections.
Analysis & Results: Improved – This section is improved but case study findings reported are still a bit repetitive. Although authors improved this section they can have a second look to make sure they represent what they are intended to share.
Also, I am not fully convinced with the author(s) position on the following comment but I am fine, if the authors and the journal editor feels this is sufficient.
- Mentioned in review.1 - Another major point for me that must be addressed is that I am truly not clear why four case studies and the two case studies analyzed separately? If the research suggests that it follows completed/constructed projects as “landscape performance” approach suggests why they are not combined? If they are different type of case studies and require different evaluation systems why they are covered in the same paper? Authors owe reader critical explanation of the analysis procedures.
Discussion & Conclusion: Improved.
Language & Formatting:
- As much as I appriciate the revised and improved language by using professional editor the paper still requires additional editing for clarity, precision, grammar, punctuation, and formatting. Please see following lines just for example issues.
References: The list is improved but formatting needs work.
- Referencing seems inconsistent in formatting. All references in the final list needs to be revised for consistency, accuracy, and style/format.
Line-by-line (sample comments):
Line 36 – What do you mean by more heat?
Line 38 - Shouldn’t this be “to meet”
Line 40 – Please check this sentence for structure and meaning.
Line 57 – Future tense is not typically recommended for such papers.
After Page.4
Pg.4 - Line.1 Do you mean “Case Study Overviews”?
Line.14 - Maps are not legible.
Line.100 and 101 in methods AHP & FCE needs proper references. Since they are first introduced here!!!
Line 117-119 - Please revise sentence. Not sure you mean through throughout or something else here. Full sentence may need revision.
Line 142- What is following data?
Line 146-147 font size look different
Line 148 “Committed to improve” or “improving”? “the” looks bolded.
Line 161 – Indent does not make sense.
Line 169 – Extra period
Line 187-188 Sentence does not make sense and grammatically problematic.
Line 211 – Spacing in these pages as well as location of figure title does not make sense in these pages (see i.e. 13 and 14)
Line 223 – “ In the following case study” which case study? What are you referring to? One or many?
Section 4.1. I am not sure the term “Verification” for case study subtitles are appropriate given that researcher assumed to be unbiased until the case study complete. I would consider taking those out from subtitles.
Please check – all tables (see such as 3 and 4) and or figures are referred in text.
Line numbers disappeared so make sure to correct the language in 4.7.1 (same for 4.7.2 Do you mean Finding & Discussion or “Finding: Discussion…”?
Section 5 Conclusion – do you mean economic, environmental, social?
- “There are certain study discussions worth noting.” Not a scholarly sentence and doesn’t mean much. Please revise.
Thank you for the revised work looking forward for a strong and successful completion of the paper.
Author Response
Thank you so much for your second review work again! Your comments really help us to further improve the study. We do hope the revised paper to be a strong and successful completion as our reviewers have suggested! Here are the responses for minor review comments as the editor required. As always, the specific track parts of revise for the paper are indicated directly through the binding tracking-change in the revised manuscript:
Please see the attached file for the detailed response to your comments.

Reviewer 4 Report
I still wonder why have the authors choose the British and American evaluation systems, it would be great to provide some short explanations.
Authors reffer to six case studies, while the fig. 10 shows comparisons of 4 cases, and the fig. 11 - comparison of 2 cases (not included at fig. 10). Why? I think all 6 cases should be given for both figures.
I would suggest to reverse a "flow" of the scheme shown at the fig. 1 - it is a common rule to procede from left to right (as at the fig. 9).
Author Response

(The authors gave the same response as above.)

Reviewer 5 Report
The revised manuscript shows improvements, but the main critical issue regards the scientific soundness. Please see the detailed report below.
Major issues
Section ‘1. Introduction’
An international scholar would expect a proper literature review that introduces the scientific work. It would be interesting report on (i) who has dealt with the same topic discussed in the paper under review, (ii) if the topic has been acknowledged as important by the international scientific community, (iii) the critical issues stressed by other scholars, and (iv) which of such critical issues the authors addressed in the manuscript.
Lines 54-56: “The coordinated development of villages and towns from the perspective of “Ecological Priority and All-Area Integrated Development” represents a new form with network-like pattern”. I would suggest the authors stress if it is a new form in China or worldwide, and if other scholars deal with similar topics.
Lines 78-80: “This all-area integrated collaborative development model can give full play to achieve resources and infrastructure sharing for peripheral towns, promoting regional linkage between various towns (As shown by Figure 7)”. It seems an authors’ personal opinion. I would suggest the authors add some references or data that support the statement.
Section ‘3. Methodology’
The content of this section is weak from a scientific point of view, because the authors do not report on any similar approach discussed in international scientific literature. Then, the contents of this section on methodology and method appears lacking scientific basis. I appreciate that the authors have added the scientific references [18] and [19] in Table 1. Reference [19] should be: “Wang, Z., Yang, B., Li, S., Binder, C. (2016). Economic benefits: Metrics and methods for landscape performance assessment. Sustainability (Switzerland), 8 (5), art. no. 424”. Please check.
Lines 193-195: “The newly composed evaluation criteria for village and town construction should be a combined use of micro and macro indicators”. Are such types of indicators described in scientific literature?
Lines 200-225: the ‘Research Methods’ are poorly discussed by the authors. Figures 8 and 9 should be clearly described by the authors. I would suggest the authors briefly report on international scientific research that used similar approaches.
Section '5. Conclusion'
I would suggest the authors clearly point out the reasons the manuscript would be relevant to international scholars.
Minor issues
Page 3 is empty.
Section ‘2. Cases Overview’
Line 36: “1.1.1. Rural development with more heat in the current China context”. I am not sure about the meaning of ‘heat’ in the title. I would suggest the authors explain what they mean with such a title.
Figure 2: the meaning of Figure 2 appears unclear given it has not been described in the text.
Line 90: “3.1. Research ideas and methods (As shown by Figure 1)”. Please consider removing “(As shown by Figure 1)” from the title. It appears superfluous. Furthermore, Figure 1 is after Figure 7: please check.
Line 214: “semi -quantitative and semi-quantitative”. Please check.
The authors refer to:
“[7]中华人民共和国住房和城乡建设部.《城市总体规划实施评估办法(试行)”;
“[5]浙江省小城镇环境综合整治行动领导小组办公室, 《浙江省小城镇环境综合整治三年行动计划》.Available online: http://www.zjjs.gov.cn/n18/n142/n147/c355062/content.html”;
I would suggest the authors also include an English translation of the references.
Author Response

(The authors gave the same response as above.)
